# Transcriptomic Landscape of Circulating Extracellular Vesicles in Heart Transplant Ischemia–Reperfusion

**DOI:** 10.3390/genes14112101

**Published:** 2023-11-18

**Authors:** SeoJeong Joo, Kishor Dhaygude, Sofie Westerberg, Rainer Krebs, Maija Puhka, Emil Holmström, Simo Syrjälä, Antti I. Nykänen, Karl Lemström

**Affiliations:** 1Translational Immunology Research Program, Transplantation Laboratory, University of Helsinki, 00014 Helsinki, Finland; seojeong.joo@helsinki.fi (S.J.); kishor.dhaygude@helsinki.fi (K.D.); sofie.westerberg@helsinki.fi (S.W.); rainer.krebs@helsinki.fi (R.K.); emil.j.holmstrom@helsinki.fi (E.H.); simo.syrjala@helsinki.fi (S.S.); antti.nykanen@helsinki.fi (A.I.N.); 2Institute for Molecular Medicine Finland FIMM, EV and HiPREP Core, University of Helsinki, 00014 Helsinki, Finland; maija.puhka@helsinki.fi; 3Heart and Lung Center, Helsinki University Hospital, University of Helsinki, 00014 Helsinki, Finland

**Keywords:** extracellular vesicles, exosomes, plasma, heart transplantation, ischemia–reperfusion injury, RNA sequencing, primary graft dysfunction

## Abstract

Ischemia–reperfusion injury (IRI) is an inevitable event during heart transplantation, which is known to exacerbate damage to the allograft. However, the precise mechanisms underlying IRI remain incompletely understood. Here, we profiled the whole transcriptome of plasma extracellular vesicles (EVs) by RNA sequencing from 41 heart transplant recipients immediately before and at 12 h after transplant reperfusion. We found that the expression of 1317 protein-coding genes in plasma EVs was changed at 12 h after reperfusion. Upregulated genes of plasma EVs were related to metabolism and immune activation, while downregulated genes were related to cell survival and extracellular matrix organization. In addition, we performed correlation analyses between EV transcriptome and intensity of graft IRI (i.e., cardiomyocyte injury), as well as EV transcriptome and primary graft dysfunction, as well as any biopsy-proven acute rejection after heart transplantation. We ultimately revealed that at 12 h after reperfusion, 4 plasma EV genes (*ITPKA*, *DDIT4L*, *CD19*, and *CYP4A11)* correlated with both cardiomyocyte injury and primary graft dysfunction, suggesting that EVs are sensitive indicators of reperfusion injury reflecting lipid metabolism-induced stress and imbalance in calcium homeostasis. In conclusion, we show that profiling plasma EV gene expression may enlighten the mechanisms of heart transplant IRI.

## 1. Introduction

Heart transplantation (HTx) remains the ultimate therapeutic option for patients with end-stage heart failure. Although outcomes have improved due to advances in immunosuppressive care and organ preservation methods, the short- and long-term survival of the allograft is negatively affected by factors such as ischemia–reperfusion injury (IRI). This inevitable complication after HTx is induced by periods of cold and warm ischemia during preservation and surgical procedure, followed by reperfusion and concomitant reoxygenation after implantation, altogether eventually causing exacerbation of cellular damage and primary graft dysfunction (PGD) [1,2]. Therefore, studies and strategies for the prevention of this phenomenon are of high clinical significance. 

Extracellular vesicles (EVs) are a heterogeneous group of nanosized membranous vesicles secreted by all cell types. EVs play a role in cell-to-cell communication in various physiological processes as they carry, e.g., proteins, carbohydrates, lipids, messenger RNAs (mRNAs), and microRNAs that can interact with target cells [3]. In the regulation of inflammation, they have now been recognized as potential biomarkers and immune mediators in organ transplantation [4].

Ischemia–reperfusion-induced EV release may worsen myocardial injury by promoting inflammation [5]. However, their role in IRI after cardiac transplantation remains poorly understood. We aimed to characterize the transcriptomic landscape of plasma-derived EVs during IRI in human HTx using RNA sequencing. We compared gene expression changes in EVs derived from the plasma of heart transplant recipients before and at 12 h after reperfusion. Furthermore, we correlated clinical data with the plasma EV transcriptome to identify EV transcripts that are strongly linked to IRI-related myocardial injury and the development of PGD.

## 2. Materials and Methods

### 2.1. Study Design

This study was designed to characterize the recipient plasma EV transcriptome during ischemia–reperfusion in human HTx. In addition, we integrated the clinical data of the recipients to identify significant correlations between the EV transcriptome and occurrences of myocardial injury and PGD at 12 h after reperfusion. The patient cohort was initially designed for a clinical trial on donor simvastatin treatment [6,7] and was conducted as a prospective, double-blind, randomized single-center study at the Helsinki University Hospital between 2010 and 2016. The study was approved by the Ethical Committee Board (permission number 358/13/03/02/2009), and it was conducted in accordance with the principles of the Declaration of Helsinki. The blood samples used in this study were collected at two timepoints: immediately before reperfusion (0 h) and at 12 h (12 h) after reperfusion. Using next-generation sequencing technology, we profiled the whole EV transcriptome in an unbiased approach to thoroughly understand expression changes within the EV transcriptomic landscape. The classification of PGD grades was made within 24 h after the transplantation, in accordance with the ISHLT guidelines [8]. For transcriptomic profiling and subsequent correlation analyses with clinical data, we initially used paired plasma samples from 41 recipients. Following stringent sequencing data control, the finally analyzed dataset consisted of paired plasma samples from 38 recipients.

### 2.2. Sample Collection and Processing

Blood samples from 41 heart transplant recipients were collected into lithium heparin BD Vacutainer blood tubes (Becton Dickinson, Franklin Lakes, NJ, USA) immediately before reperfusion (0 h) and at 12 h after reperfusion (12 h) during the HTx procedure. Blood was centrifuged at 1600× *g* for 10 min at RT. Plasma was transferred to cryotubes (Nunc Thermo Scientific, Waltham, MA, USA) and stored at −80 °C. Upon analysis, 900 μL of thawed plasma was diluted at a 1:1 ratio with 0.22 μm filtered PBS and filtered with 0.8 μm Millex AA syringe filters (Millipore Sigma, Burlington, MA, USA). To isolate EV, diluted plasma was ultracentrifuged using SW 55 Ti rotor and Polypropylene Thickwall tubes (Beckmann-Coulter, Brea, CA, USA) at 40,000 rpm at +4 °C for 90 min. EV pellets were washed with 2 mL of PBS and ultracentrifuged, resuspended in 100 μL of filtered PBS, and stored in DNALobind tubes (Eppendorf, Hamburg, Germany). To eliminate heparin residues, heparinase I (New England Biolabs, Ipswich, MA, USA) was added to the EV resuspension (0.1 U/μL) and incubated at 37 °C for 1 h. RNA was isolated from EVs by using a Qiagen miRNeasy micro kit (Qiagen, Hilden, Germany). The quality and concentration of EV RNA were measured using the Bioanalyzer Pico kit (Agilent, Santa Clara, CA, USA). 

### 2.3. Transmission Electron Microscopy and Nanoparticle Tracking Analysis

To examine the presence and morphology of isolated EV, representative samples were prepared for transmission electron microscopy (0 h, *n* = 6 and 12 h, *n* = 5) [9]. To analyze the concentration and size distribution of isolated EV, 20 μL of the representative eluted EV samples (0 h, *n* = 5, and 12 h, *n* = 8) were diluted in Dulbecco’s PBS and analyzed on NanoSight LM14C (Nanosight, Salisbury, UK) and Zetaview PMX-120 model (ParticleMetrix, Ammersee, Germany) equipped with a blue laser (404 nm and 488 nm, respectively) and a high sensitivity CMOS camera. EV size and concentration data were obtained with the NanoSight instrument by recording 90 s of 3 videos, and EV size and concentration data analyzed with Zetaview were captured by scanning 11 positions with 30 frames and 12.41 μS/cm sensed conductivity. To determine the concentration and the size of isolated EV, data obtained were analyzed using NTA software 3.0 and ZetaView software (Version 8.05.12 SP2).

### 2.4. RNA Sequencing 

EV RNA samples from 17 recipients were sequenced on Illumina HiSeq 2500 and samples from 21 recipients were sequenced on Illumina NovaSeq 6000 (Illumina, San Diego, CA, USA). Pair-ended cDNA libraries were prepared using SMARTseqv4 Ultra Low Input RNA kit (Takara Bio Inc., Kusatsu, Shiga, Japan). One hundred picograms of amplified cDNA were tagmented and indexed using the Nextera XT DNA library prep kit (Illumina). Finally, cDNA libraries were sequenced using a sequencing v4 kit (Illumina) to obtain at least 15 million reads on the HiSeq and 25 million reads on the NovaSeq platform. 

### 2.5. Bioinformatics

RNAseq raw data were processed with the Trimmomatic software (version 0.39) [10] and were aligned to the Human Genome (GRCh38) with the STAR software (version 2.7.10a) [11]. Gene expression read counts were calculated by using the Rsubread software (version 2.16.0) [12]. Sequencing data were normalized with the TMM normalization method prior to the statistical analysis to avoid differences between sequencing platforms and batches. Raw counts, normalized CPM, and RPKM values are given in Appendix A. Outliers were identified by generating a principal component analysis (PCA) plot from counts per million (CPM) values for each gene in the sample. Analyses such as PCA and unsupervised hierarchical clustering were performed with the ropls (version 1.34.0) [13] and factoextra R packages (version 1.0.7). Differential gene expression testing was performed using edgeR (version 4.0.1) [14]. Protein-coding genes presenting absolute log-change greater than 1.5 and *q*-value (FDR-adjusted *p*-value) less than 0.05 were considered to be significant. In addition, the R package CombiROC (version 0.3.4) and pROC (version 1.18.5) were used to calculate the power of discrimination for 1317 differentially expressed genes (DEGs). Heatmaps of dDEGs were visualized by the pheatmap R package (version 1.0.12) [15]. Functional annotations of DEGs were investigated with the clusterProfiler (version 4.4.4) using its functions enrichGO and enrichKEGG. Enriched terms with a *q*-value (FDR-adjusted *p*-value) of less than 0.05 were considered significant. Outputs of enrichment analyses were visualized using the dotplot function in the clusterProfiler R package [16]. We summarized and reduced the redundant enriched gene ontology terms by using the Revigo-based [17] R package RRVGO (version 1.10.0) [18]. Summarized terms were visualized using the scatterplot function included in the same package. To examine the protein network of the genes, we used the STRING analysis tool (https://string-db.org/, version 12.0, last accessed on 2 October 2023) [19]. Furthermore, we used Web-based cell-specific enrichment analysis (WebCSEA, https://bioinfo.uth.edu/webcsea/, last accessed on 6 October 2023) [20] to elucidate the cell types that may be associated with the significant 1317 protein-coding genes found at 12 h post-reperfusion when compared to 0 h. This tool enables the enrichment of systematic gene sets for human tissue cell-type expression signatures. The most significant cell types were identified and visualized based on the −log10 fold change combined with the *p*-value. For correlation analysis of high-sensitivity plasma troponin I level (TnI), PGD, and AR with EV transcriptome, we used the Wilcox rank sum test. The final set of significant genes was determined with a *p*-value cutoff of less than 0.005.

## 3. Results

### 3.1. Recipient Characteristics

Plasma EV RNA was extracted from 41 heart transplant recipients, from samples taken immediately before and at 12 h after reperfusion and then subjected to next-generation RNA sequencing. Three recipients were excluded due to the low RNA quality of their samples. Of the remaining 38 recipients, the median age was 60 (range 27–67), and 9 (23.7%) of them were female. The most common cause for HTx was dilated cardiomyopathy (46.2%). A total of 26.2% of patients were bridged to transplantation with mechanical circulatory support (Table 1). A CONSORT flowchart describes the study flow (Figure 1).

### 3.2. EV Characterization

Plasma EVs showed a typical cup-shaped morphology in the electron microscopic images (0 h, *n* = 6 and 12 h, *n* = 5) (Figure 2A,B). Nanoparticle tracking analysis revealed that the size distribution of isolated EVs from 0 h (*n* = 5) and 12 h (*n* = 8) samples ranged within approximately 50 nm to 1355 nm and 50 nm to 1095 nm, respectively. The plasma EV concentration was similar at both timepoints (Figure 2C,D). 

### 3.3. Reperfusion Altered mRNA Expression Profiles of Recipient Plasma EV

Unsupervised PCA showed distinct plasma EV transcriptomic profiles at 12 h after heart transplant reperfusion compared to 0 h before reperfusion (Figure 3A,B). Raw counts, normalized CPM, and RPKM values are given in Appendix A. DEG analysis, with a cut-off of log-fold change greater than 1.5 and a *q*-value less than 0.05, revealed a total of 1317 (3.4%) protein-coding genes. Among these, 25.6% of the genes were found to be upregulated, while 74.4% were downregulated (Figure 3C and Appendix A). We additionally conducted ROC analysis to assess different powers of discrimination for DEGs (Appendix A). When applied minimum specificity of 80% and sensitivity of 80% for high-power discrimination, 27% of DEGs (357 out of 1317) at 12 h passed the cutoff criteria (Appendix A). These genes exhibited similar biological functions to those we observed in pathway analyses when using 1317 DEGs (Appendix A). 

### 3.4. Biological Functions Related to the Differentially Expressed Genes after Reperfusion

To gain a better understanding of the impact of heart transplant reperfusion injury on the plasma EV transcriptome, we investigated the 1317 DEGs by enriching biological annotations. Firstly, we performed gene ontology enrichment over-representation analysis to explore relevant biological processes. Upregulated genes at 12 h after reperfusion were mainly linked to cell division, leukocyte-mediated immunity, and metabolic processes (Figure 4A), whereas downregulated genes were related to extracellular structure, tissue development, and cell proliferation (Figure 4B). To refine and reduce possible redundant enriched gene ontology terms, we summarized those terms based on a semantic similarity score by the Relevance method [21]. The calculated scores thus yielded more precise terms related to biological processes. We found that upregulated genes were related to leukocyte-mediated immunity, acute-phase response, regulation of coagulation, lipid metabolic processes, and mitosis (Appendix A), while downregulated genes were linked to complement activation, cell chemotaxis, wound healing, response to oxygen levels, organ development, and collagen processes (Appendix A). Next, we conducted a KEGG pathway enrichment analysis to explore possible biological pathways related to the DEGs. In alignment with the findings in the gene ontology analysis, upregulated genes were associated with cytokine receptor interaction, cell cycle, natural-killer-cell-mediated cytotoxicity, retinol metabolism, antigen processing and presentation, allograft rejection, and complement and coagulation cascades (Figure 4C and Appendix A). Downregulated genes were related to cytokine interaction, cell signaling pathways (e.g., PI3K-Akt, AGE-RAGE, Rap1, Hippo, and TGF-β), focal adhesion, complement and coagulation cascades, and ECM–receptor interaction (Figure 4D and Appendix A). Lastly, we performed WebCSEA analysis to get a better insight into the possible origin of EV transcripts, based on the tissue cell-specific signatures. The results suggested that DEGs at 12 h after reperfusion showed various immune cell and organ tissue cell-type signatures in the overall organ system (Figure 4E). Since we analyzed plasma samples from heart transplant recipients, we specifically investigated whether the DEGs were related to heart tissue. The results suggested that DEGs at 12 h after reperfusion showed cell-type-specific signatures associated with fibroblasts, stromal cells, smooth muscle cells, pericytes, mesenchymal stem cells, epicardial adipocytes, endocardial cells, endothelial cells, and Schwann cells in the heart, based on the Tabula Sapiens database [22] (Figure 4F and Appendix A). Although Tabula Sapiens contains hepatocytes in the heart tissue cell signatures, we excluded hepatocytes from our results due to their irrelevance to heart histology. In addition, with detailed analysis by separating upregulated and downregulated genes, we observed heart-related immune cells were mainly upregulated and cell structure component genes were downregulated (Appendix A). 

### 3.5. Post-Reperfusion EVs Carry More Adaptive Immunity-Related Genes but Less Extracellular Matrix Component Protein-Coding Genes

Furthermore, we hypothesized that the genes with higher log-fold changes may represent the most significant biological functions related to those genes. Therefore, we filtered the DEGs based on the log fold-change and selected 50 upregulated and 50 downregulated genes for STRING analysis, rather than focusing on genes related to the most significantly enriched GO and pathway terms to avoid biased interpretation of established gene networks. STRING analysis was performed to explore the protein–protein network of DEGs, as well as relevant biological annotations of these genes. The most upregulated 50 genes were associated with retinol metabolism, complement system, and graft-versus-host disease, sharing a gene set related to allograft rejection (Figure 5A). Downregulated genes showed well-established protein–protein networks, and the abundantly enriched biological annotations were related to extracellular matrix organization, including collagen-containing extracellular matrix, and organ development (Figure 5B).

### 3.6. Correlation of EV Transcriptome with Cardiomyocyte Injury

Our previous study has reported that plasma high-sensitivity troponins are highest at 12 h in recipients who had severe left ventricular PGD [23]. Therefore, we investigated all EV transcripts expressed at 12 h to identify significant correlations between the EV transcriptome and cardiomyocyte injury, defined by 12 h plasma level of high-sensitivity TnI. We found that 48 protein-coding genes were significantly correlated with myocardial injury (Figure 6A and Table 2), and they exhibited correlations in their expression with each other (Figure 6B). In addition, of the 48 genes correlated to TnI plasma levels, 8 were also found among the DEG after reperfusion, suggesting their expression patterns are significantly affected after reperfusion (Appendix A).

### 3.7. Association of EV Transcriptome and Outcomes after Heart Transplantation

We next compared the plasma EV transcriptome of recipients without PGD (66%) with those who developed PGD of any grade (34%). The result revealed that 3194 protein-coding genes were related to the higher incidence of PGD when applying a *p*-value of less than 0.05. Generally, these genes that showed correlations with PGD (*p* < 0.05, n = 3194) were associated with lipid metabolism, acute-phase response, glucuronidation, TNF signaling pathway, and complement and coagulation cascades (Appendix A). We found 146 protein-coding genes when using a more stringent *p*-value cut-off (*p*-value < 0.005) (Table 3) and 30 protein-coding genes when applying a *p*-value of less than 0.001. Most of these 30 genes were upregulated compared to recipients without PGD (Appendix A). To investigate whether the gene expression of these 146 genes was affected by reperfusion, we compared the genes with 1317 DEGs at 12 h after reperfusion. We found 25 genes that were also present in the DEGs after reperfusion, and their biological functions were related to retinol metabolism, amyloidosis, bone marrow cells, plasma cells, liver, and digestive gland (Appendix A). However, no significant differences in donor characteristics were observed between the recipient PGD groups (Appendix A). Given that cardiomyocyte damage is a prominent manifestation of PGD after HTx, we cross-checked our correlation analyses, and identified four protein-coding genes (*ITPKA*, *DDIT4L*, *CD19*, and *CYP4A11*) that were consistently observed (Figure 6C and Table 4). These genes showed correlations in expression among each other (Figure 6D) and three of them showed upregulation in the recipient group with PGD after HTx (Figure 6E). Although reperfusion injury may have a greater impact on the early outcome (i.e., PGD) than on long-term outcomes such as acute rejection, we investigated the potential relationship between genes related to reperfusion injury and biopsy-proven acute rejection. We examined differences in the EV transcriptome at 12 h based on the recipients’ IV-treatment history within 30 days after HTx and biopsy-proven AR diagnosis history within the first year after HTx. Among the 38 recipients, 9 received IV treatment within 30 days after HTx, while 27 did not (2 not available). Additionally, 24 recipients developed any biopsy-proven AR within the first year after HTx, and 14 recipients remained stable. We found 100 genes that showed significantly different expression levels based on the history of IV treatment within 30 days and another 100 genes associated with biopsy-proven AR within the first year, respectively (*p* < 0.01). Moreover, 31 and 42 genes were found in each analysis when applying a *p*-value cutoff of 0.005 (Appendix A). However, we did not find any significantly enriched pathways and gene ontology terms using biopsy-proven AR prediction genes.

## 4. Discussion

In this study, we provide a detailed analysis of the transcriptomic landscape of plasma-derived EVs following IRI in human HTx. While transcriptomic signatures of IRI in experimental settings have been well established using bulk RNAseq, the whole transcriptomic profile of plasma EVs in clinical settings has been unexplored. We uncovered 1317 EV transcripts that were significantly altered during heart transplant IRI, including upregulation of genes related to immune activation and metabolism mediated by cytochrome P450 (CYP) and downregulation of genes associated with cell proliferation, extracellular matrix, smooth muscle contraction, and TGF-β signaling pathway. In addition, EV transcripts were linked to cardiomyocyte injury and PGD as 48 protein-coding EV transcripts were associated with higher levels of the cardiomyocyte injury marker TnI and 30 EV genes with the development of PGD after HTx. Importantly, four transcripts—*ITPKA*, *DDIT4L*, *CD19,* and *CYP4A11—*were closely associated with both cardiomyocyte injury and PGD, indicating that these protein-coding genes may have important roles in regulating heart transplant IRI.

A recent systematic review of bulk RNA analyses of myocardial tissue suggests that the transcriptomic hallmarks of IRI include increased mRNA expressions related to response to stress, cell proliferation, inflammatory response, and cell pathway (i.e., TNF, NF-kB, IL-17, MAPK, TLR, and NOD-like signaling pathways), while downregulated genes were associated with cytoskeletal structures, cell development, cell survival, ion channels, and cAMP signaling pathway [24]. Our findings showed that recipient plasma EV transcripts were associated with upregulated genes related to inflammation, T-cell chemotaxis, and cytokine interactions, while downregulated genes were linked to ion transport and extracellular matrix (Appendix A). Interestingly, we found that circulating EVs contain gene transcripts related to allograft rejection already at 12 h after reperfusion, along with transcripts associated with cell proliferation and metabolism. Moreover, WebCSEA analysis focused on the heart tissue database suggested that DEGs at 12 h after reperfusion were related to a broad range of tissue structural components, such as fibroblasts, stromal cells, smooth muscle cells, pericytes, epicardial adipocytes, and others related to heart tissue. Nevertheless, our pathway analyses and tissue-specific cell type enrichment analyses suggested that circulating EVs after reperfusion may be involved in or reflect the regulation of cellular structure, hence affecting tissue remodeling and repair in the cardiovascular system after reperfusion. Collectively, our results may help to understand how the cellular structure disruption and inflammation after IRI contribute to the development of PGD after HTx.

Upon reperfusion of the heart transplant, oxygen radicals may cause tissue damage by interacting with polyunsaturated fatty acids and by the formation of lipid peroxides and hydroperoxides. Oxygen free radicals induce the release of platelet-activating factors by endothelial cells and, thereby, aggravate neutrophil-mediated immune responses. Consequently, elevated proinflammatory signals and activation of downstream signaling pathways may ultimately result in the development of parenchymal fibrosis and decreased cardiac function. Given that, we speculate that lipid metabolism and lipid-metabolism-induced stress are key factors for understanding the mechanism of IRI. For example, polyunsaturated fatty acids such as arachidonic acid and linoleic acid are well-known to participate in the induction of endothelial dysfunction, vascular tone, calcium mobilization, and oxidative stress [24,25,26,27,28,29]. Especially, arachidonic acid is an essential fatty acid and precursor of prostaglandins, thromboxanes, and leukotrienes, which are important players in endothelial dysfunction [30]. Of the four genes that showed a moderate correlation with both cardiomyocyte injury and PGD, CYP4A11 has been shown to have a role in lipid metabolism, reactive-oxygen-species-induced lipid peroxidation, and inflammation in non-alcoholic fatty liver disease [31]. There is growing evidence suggesting that CYP enzymes play a pivotal role in the cardiovascular system [32]; for example, polymorphisms in CYP4A11 have associations with coronary artery diseases [33]. Furthermore, *DDIT4L*, another one of the four genes highlighted in the results, promotes cardiomyocyte cell death by inhibiting the mTOR signaling pathway under hypoxia/reoxygenation setting [34] and increases oxidized LDL-induced cytotoxicity [35], which may indicate the relation between oxidative stress and lipid metabolism. Furthermore, *DDIT4L* expression is reported in pathological cardiac hypertrophy, and overexpression of DDIT4L increases autophagy and causes mild systolic dysfunction [36].

Next, we focused on the calcium signaling pathway. During IRI, calcium overload can cause cardiac systolic dysfunction, because calcium ions play an essential role in cardiac action potential [37]. We found both *ITPKA* and *CD19* may participate in the regulation of calcium transport in cardiomyocytes and the PI3K signaling pathway. Although CD19 is well known as a marker for B cells, we focused on the role of CD19 and B-cell receptor pathway cascades that participate in calcium homeostasis [38]. BCR-induced calcium transport has a connection with G-protein-coupled receptors, which are closely related to cardiovascular health [39], and enhanced BCR signaling under deficiency of CD19 is related to PI3K activity [40], and changes in myocardial B-cell population may affect myocardial growth and contractility [41]. Downregulation of calcium signaling pathway-related transcripts may result in impaired cardiac contractibility and remodeling due to dysregulation of excitation–contraction coupling [42]. Altogether, an imbalance in calcium homeostasis and activation of downstream signaling pathways may significantly affect the outcome of a heart transplant by altering the metabolism and result in mitochondrial dysfunction, inflammation, and reduced heart function.

One limitation of this study is the relatively small cohort size of 38 recipients. Additionally, validation of RNAseq data by qRT-PCR was not performed due to a limited volume of samples. Revalidation of the data in a larger cohort using both RNAseq and qPCR could further enhance our understanding of the mechanistic connection between EV and IRI and the role of EV in HTx. Moreover, we cannot differentiate transplantation-associated IRI from normal IRI, as well as alloimmune response reflected in circulating EVs after reperfusion. In addition, comparing blood samples from the coronary sinus and peripheral blood may provide enhanced specificity of EV analysis related to cardiac conditions. However, we did not consider this when initiating the original randomized control trial. Future studies using external cohorts and comparisons with samples taken from recipients before transplantation and during ischemia may enable us to differentiate the effects of ischemia-induced and reperfusion-induced injuries on the EV transcriptome.

## 5. Conclusions

In conclusion, we provide a comprehensive insight into the whole transcriptomic profile of plasma EVs from heart transplant recipients after reperfusion. In addition, we highlight four transcripts that EVs carry after reperfusion and suggest their clinical relevance both in cardiomyocyte injury and in PGD after HTx. Altogether, our findings suggest that EV may provide important information about heart transplant IRI. In addition to our analysis focused on protein-coding genes in plasma EVs, investigating the expression profiles of long non-coding RNAs, pseudogenes, and microRNAs may provide a deeper understanding of the EV biomarker capabilities or intricate roles of plasma EVs as gene expression modulators in IRI. Additional in-depth studies should be carried out to further elucidate the possible tissue sources of the circulating EVs and their roles in IRI and allograft damage. Ultimately, the profiling of EV transcriptome may be utilized as a novel approach for the development of non-invasive biomarkers for PGD after HTx.

## Figures and Tables

**Figure 1 genes-14-02101-f001:**
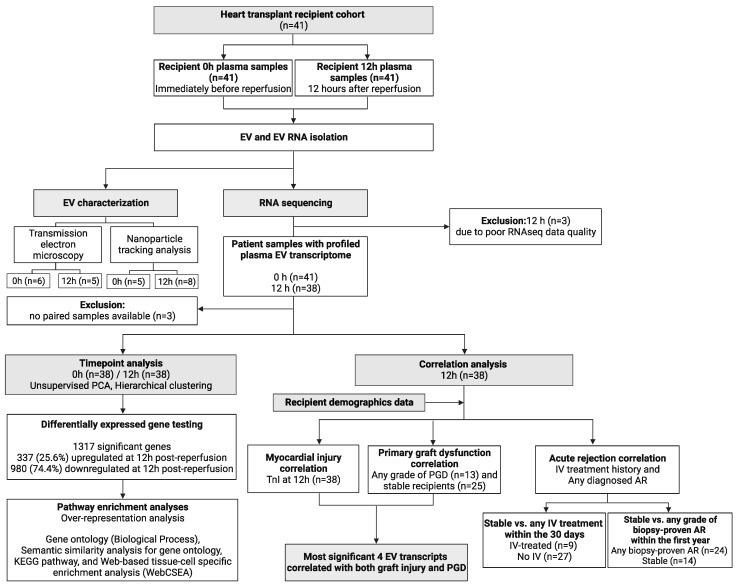
CONSORT-type workflow chart of the study. Plasma EV transcriptomes before reperfusion and at 12 h after reperfusion of the heart transplant were compared with each other. Only the EV transcriptome of plasma samples taken at 12 h after reperfusion was correlated with the clinical data of the heart transplant recipient. Figure created with BioRender.com (accessed on 14 November 2023).

**Figure 2 genes-14-02101-f002:**
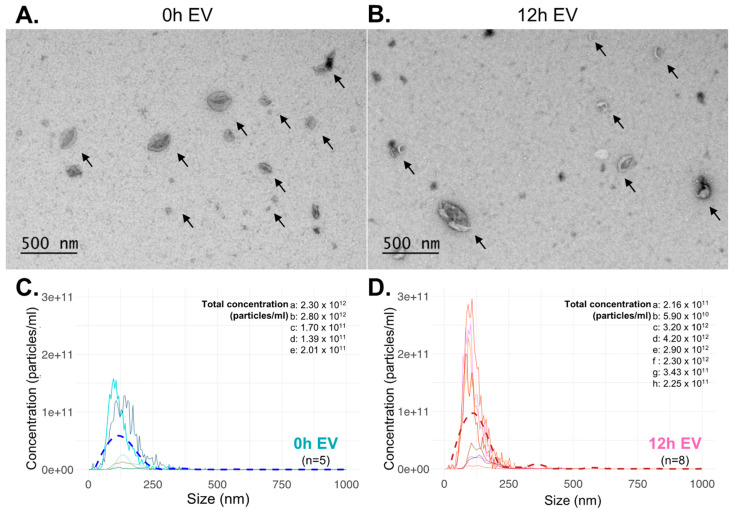
Characterization of isolated EVs. Transmission electron microscopy and nanoparticle tracking analysis were performed to confirm and examine the presence, morphology, size distribution, and concentration of recipient plasma EVs. (**A**) Transmission electron microscopy of plasma EVs at 0 h and (**B**) at 12 h after reperfusion of the heart. Arrows indicate the typical cup-shaped morphology of EVs. (**C**) Concentration (particles/mL) and (**D**) size distribution data were obtained from nanoparticle measurement equipment (NanoSight and ZetaView); the dashed line in each plot was plotted by geom_smooth function in R, using the LOESS method to represent the trend of concentration values throughout the samples according to the size of the samples.

**Figure 3 genes-14-02101-f003:**
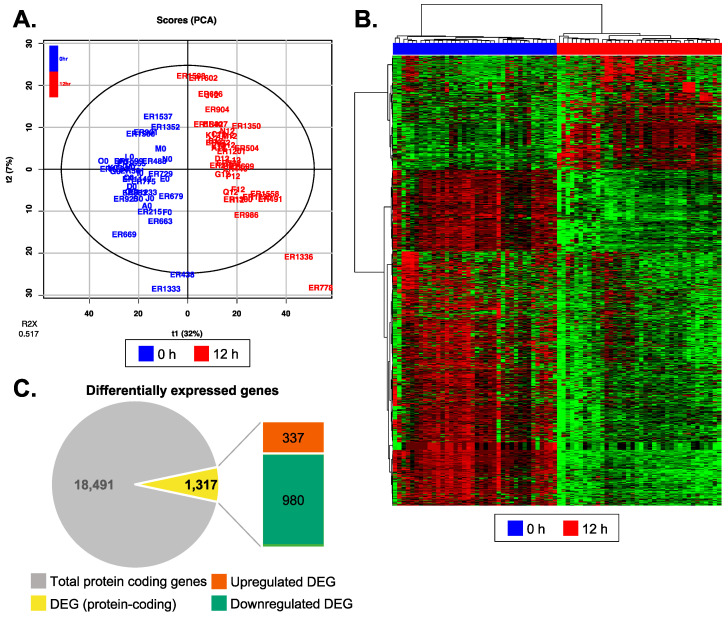
Differentially expressed genes at 12 h post-reperfusion compared to pre-reperfusion. (**A**) Unsupervised principal component analysis showed that different timepoint (blue, 0 h; red, 12 h) samples have distinct EV transcriptomic profiles. (**B**) Hierarchical clustering analysis confirmed clear differences in the gene expression patterns between 0 h and 12 h groups. (red, upregulated; green, downregulated) (**C**) The number of differentially expressed genes at 12 h after reperfusion is illustrated in a pie chart format.

**Figure 4 genes-14-02101-f004:**
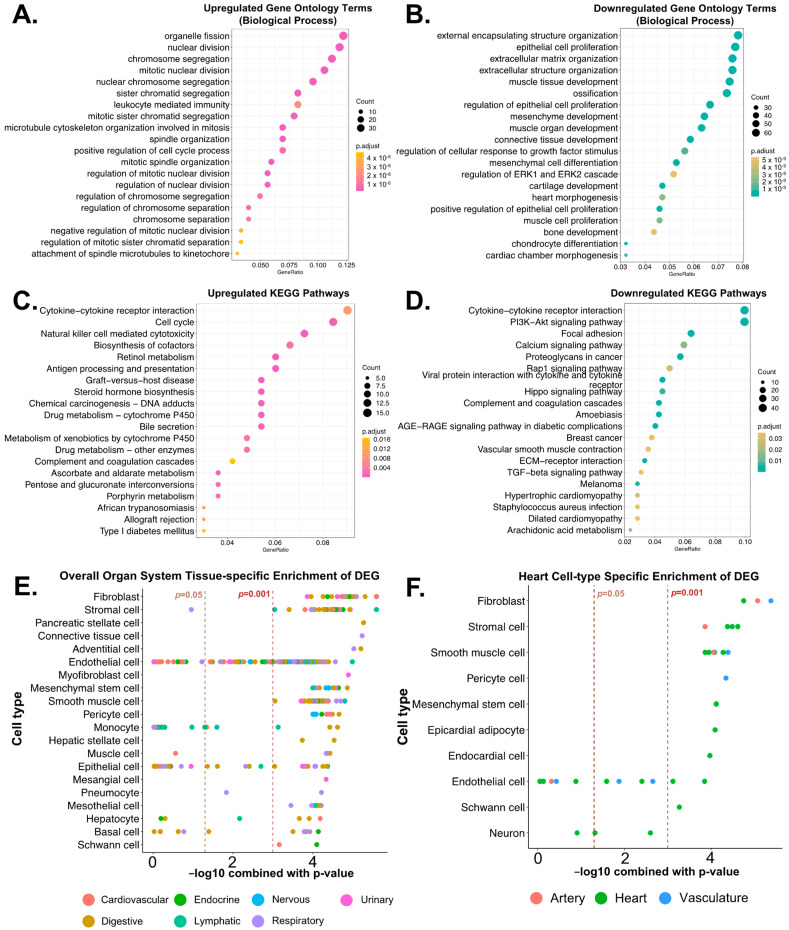
The biological relevance of differentially expressed genes found at 12 h post-reperfusion using pathway enrichment analyses. The 1317 differentially expressed genes (DEGs) were analyzed to explore their biological functions by enriching relevant terms with different databases. Focusing on biological processes, gene ontology enrichment revealed terms associated with (**A**) upregulated DEGs and (**B**) downregulated DEGs. Enriched pathways based on the KEGG database suggested relevant pathways of (**C**) upregulated and (**D**) downregulated DEGs. (**E**) WebCSEA results to visualize overall organ system-specific cell signatures, and (**F**) WebCSEA results to visualize heart tissue cell-specific signatures. Since our patient cohort consists exclusively of cardiac transplant recipients, we focused primarily on heart-related genes.

**Figure 5 genes-14-02101-f005:**
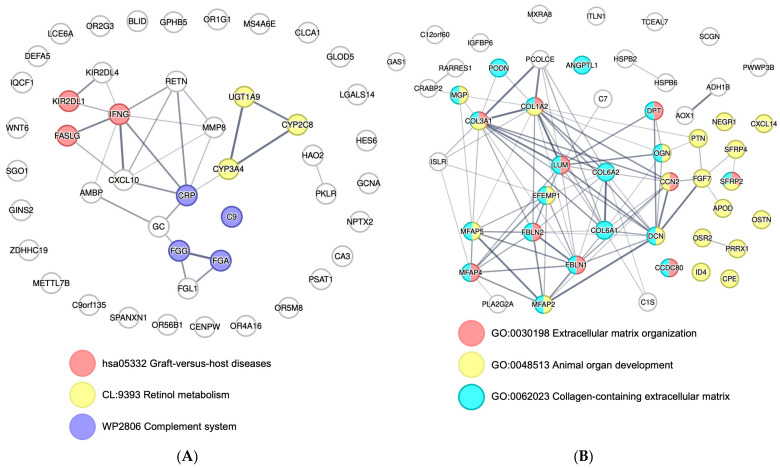
Protein network analysis of the top 50 most significantly upregulated and downregulated genes at 12 h post-reperfusion. Probable protein–protein networks and biological functions of the most significant (**A**) 50 upregulated and (**B**) 50 downregulated genes were analyzed based on the STRING online database. Edges represent protein–protein associations based on curated databases, text mining, and experimentally determined interactions. Edges are connected with lines whose thickness represents the confidence of the network association.

**Figure 6 genes-14-02101-f006:**
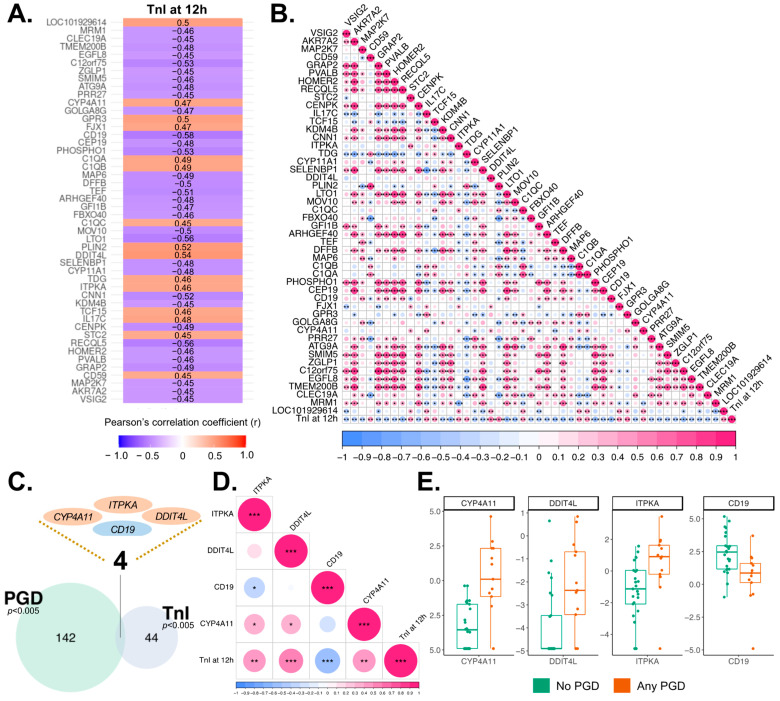
EV transcripts that showed significant correlations with clinical outcomes. (**A**) Forty-eight genes showing correlations between the EV transcriptome and the myocardial injury marker; high-sensitivity troponin I, measured at 12 h after reperfusion (*p* < 0.005). (**B**) Correlations among the expression levels of these 48 genes (*p* < 0.005). (**C**) Four genes significantly correlated with both TnI and PGD (*p*-value < 0.005). (**D**) Correlations of expression levels among the four genes. (**E**) Expression levels of the four genes (*p* < 0.005). *p*-values in B and D are denoted with asterisk marks, * *p* < 0.05, ** *p* < 0.005, *** *p* < 0.001.

**Table 1 genes-14-02101-t001:** Heart transplant recipient characteristics.

	Total (n = 38)	No PGD (n = 25)	PGD (n = 13)	*p*
Female, n (%)	9 (23.7%)	7 (28%)	2 (15.4%)	0.46
Sex mismatch, n (%)	2 (5.3%)	2 (8%)	0 (0%)	0.54
Age, median, y (range)	59 (27–67)	60 (27–67)	58 (46–66)	0.46
Weight, median, kg (range)	82 (41–120)	84 (41–112)	77 (62–120)	0.58
BMI, median, kg/m^2^ (range)	27.5 (15.6–36.3)	27.6 (15.6–36.3)	25.7 (21.5–36.2)	0.66
Panel reactive antibodies (PRA)				
PRA I, % (range)	0 (0–60)	7 (0–60)	0 (0–20)	0.14
PRA II, % (range)	0 (0)	0 (0)	0 (0)	0.34
Original diagnosis, % of known				0.16
Amyloidosis	1 (2.6%)	0 (0%)	1 (7.7%)	
Dilated cardiomyopathy	19 (50%)	13 (52%)	6 (46.2%)	
Hypertrophic cardiomyopathy	0 (0%)	0 (0%)	0 (0%)	
Ischemic cardiomyopathy	2 (5.3%)	0 (0%)	2 (15.4%)	
Congenital	1 (2.6%)	1 (4%)	0 (0%)	
End-stage coronary disease	8 (21.1%)	7 (28%)	1 (7.7%)	
Myocarditis	3 (7.9%)	2 (8%)	1 (7.7%)	
Sarcoidosis	1 (2.6%)	1 (4%)	0 (0%)	
Other	3 (7.9%)	1 (4%)	2 (15.4%)	
Chronic illnesses, % of known				
Hypertension	8 (23.5%)	6 (26.1%)	2 (18.2%)	1.00
Diabetes mellitus (I or II)	8 (26.7%)	6 (30%)	2 (20%)	0.68
Coronary artery disease	10 (29.4%)	9 (39.1%)	1 (9.1%)	0.11
Peripheral vascular disease	0 (0%)	0 (0%)	0 (0%)	1.00
Malignancy	2 (5.3%)	2 (8.7%)	0 (0%)	0.53
Prior stroke	7 (20.6%)	6 (26.1%)	1 (9.1%)	0.38
Prior heart surgery	8 (21.6%)	5 (20%)	3 (25%)	1.00
Prior sternotomy	10 (29.4%)	6 (26.1%)	4 (36.4%)	0.69
Organ-specific parameters prior to HTx				
EF, % (range)	20 (10–50)	22 (10–50)	20 (15–50)	0.89
PVR, Wood (range)	2 (1–6)	3 (1–6)	2 (1–4)	0.06
TPG, mmHg (range)	8 (2–17)	10 (2–17)	7 (3–11)	0.12
sPAP, mmHg (range)	39 (22–62)	43 (22–62)	37 (27–62)	0.39
dPAP, mmHg (range)	21 (8–39)	21 (8–39)	18 (9–33)	0.45
mPAP, mmHg (range)	28 (13–43)	28 (13–43)	26 (15–41)	0.78
Absolute FEV1, L/min (range)	2 (1–3)	2 (1–3)	2 (2–3)	0.30
Relative FEV1, % (range)	65% (14–96)	65% (14–96)	65% (55–85)	0.56
P-Bilirubin, µmol/L (range)	12 (5–46)	11 (5–46)	13 (6–40)	0.98
Pre-op GFR, mL/min/1.73 m^2^ (range)	51 (26–128)	48 (26–120)	51 (34–128)	0.71
Smoking, n (%)				0.44
Current	1 (2.6%)	0 (0%)	1 (7.7%)	
Former	13 (34.2%)	8 (34.8%)	5 (41.7%)	
Never	22 (62.9%)	15 (65.2%)	7 (58.3%)	
History unknown; only current known	3 (7.9%)	2 (8%)	1 (7.7%)	
Preoperative VAD, n (%)				0.40
Continuous flow	3 (7.9%)	3 (12%)	0 (0%)	
Pulsatile flow	3 (7.9%)	1 (4%)	2 (15.4%)	
Preoperative ECMO, n (%)	4 (10.5%)	3 (13.6%)	1 (8.3%)	1.00
Time on organ waiting list, median days (range)	117 (1–1020)	120 (1–840)	60 (2–1020)	0.77
Graft ischemia, median min (range)				
Cold	105 (7–165)	110 (7–165)	73 (9–149)	0.54
Warm	77 (30–120)	78 (30–117)	75 (40–120)	0.65
Total	160 (58–265)	170 (58–238)	156 (80–265)	0.94
Perfusion support	80 (40–186)	80 (40–125)	80 (45–186)	0.28
Nitric oxide	22 (57.9%)	11 (44%)	11 (84.6%)	0.004
RBC transfusions during Tx, units	4 (0–17)	3 (0–10)	4 (1–17)	0.07
Recipient EBV	34 (94.4%)	22 (91.7%)	12 (100%)	0.54
Recipient CMV	31 (83.8%)	22 (88%)	9 (75%)	0.37
CMV prophylaxis	35 (94.6%)	22 (91.7%)	13 (100%)	0.53
Primary graft dysfunction, n (%)				
No PGD	25 (65.8%)	25 (100%)	0 (0%)	
Any PGD	13 (34.2%)	0 (0%)	13 (100%)	
Mild PGD	2 (5.3%)	0 (0%)	2 (15.4%)	
Moderate PGD	6 (15.8%)	0 (0%)	6 (46.2%)	
Severe PGD	4 (10.5%)	0 (0%)	4 (30.8%)	
RV-PGD	1 (2.6%)	0 (0%)	1 (7.7%)	
Myocardial injury marker, median (range)				
TnI at 12 h after reperfusion	90,434 (22,757–50,000)	60,640 (22,757–35,405)	287,445 (28,711–50,000)	0.008
TnT at 12 h after reperfusion	6805 (1565–48,650)	5462 (1565–18,470)	21,040 (3356–48,650)	0.007
CPK-MB at 12 h after reperfusion	228 (51–600)	156 (51–431)	390 (95–600)	0.001

Values are given as median with range. For numeric data, statistical significance was examined by using a two-tailed *t*-test with non-equal variance, and the significance of binary data and categorical data between two or more groups was examined by Fisher’s exact test (BMI, body mass index; PRA, panel reactive antibodies; EF, ejection fraction; PVR, pulmonary vascular resistance; TPG, transpulmonary gradient; sPAP, systolic pulmonary arterial pressure; dPAP, diastolic pulmonary arterial pressure; mPAP, mean pulmonary arterial pressure; FEV1, forced expiratory volume in first second; GFR, glomerular filtration rate; VAD, ventricular assist device; ECMO, extracorporeal membrane oxygenation; RBC, red blood cell; Tx, transplantation; EBV, Epstein–Barr virus; CMV, cytomegalovirus; PGD, primary graft dysfunction; RV-PGD, right ventricular primary graft dysfunction; TnI, high-sensitivity troponin I; TnT, high-sensitivity troponin T; CPK-MB, creatine phosphokinase-MB).

**Table 2 genes-14-02101-t002:** 48 protein-coding genes that are highly related to TnI plasma levels at 12 h after reperfusion.

Symbol	Description	Coefficient r	*p*-Value
DDIT4L	DNA damage inducible transcript 4 like	0.54	4.99 × 10^–4^
PLIN2	Perilipin 2	0.52	8.25 × 10^–4^
GPR3	G-protein-coupled receptor 3	0.50	1.26 × 10^–3^
LOC101929614	Proline-rich receptor-like protein kinase PERK10	0.50	1.43 × 10^–3^
C1QB	Complement C1q B chain	0.49	1.71 × 10^–3^
C1QA	Complement C1q A chain	0.49	1.92 × 10^–3^
IL17C	Interleukin 17C	0.48	2.44 × 10^–3^
FJX1	Four-jointed box kinase 1	0.47	2.93 × 10^–3^
CYP4A11	Cytochrome P450 family 4 subfamily A member 11	0.47	2.94 × 10^–3^
TDG	Thymine DNA glycosylase	0.46	3.42 × 10^–3^
ITPKA	Inositol-trisphosphate 3-kinase A	0.46	3.65 × 10^–3^
TCF15	Transcription factor 15	0.46	3.94 × 10^–3^
CD59	CD59 molecule (CD59 blood group)	0.45	4.24 × 10^–3^
STC2	Stanniocalcin 2	0.45	4.54 × 10^–3^
C1QC	Complement C1q C chain	0.45	4.96 × 10^–3^
VSIG2	V-set and immunoglobulin domain containing 2	–0.45	4.98 × 10^–3^
PRR27	Proline-rich 27	–0.45	4.70 × 10^–3^
KDM4B	Lysine demethylase 4B	–0.45	4.64 × 10^–3^
EGFL8	EGF-like domain multiple 8	–0.45	4.56 × 10^–3^
AKR7A2	Aldo-keto reductase family 7 member A2	–0.45	4.53 × 10^–3^
ZGLP1	Zinc finger GATA-like protein 1	–0.45	4.36 × 10^–3^
MAP2K7	Mitogen-activated protein kinase kinase 7	–0.45	4.33 × 10^–3^
CLEC19A	C-type lectin domain containing 19A	–0.45	4.21 × 10^–3^
HOMER2	Homer scaffold protein 2	–0.46	4.09 × 10^–3^
MRM1	Mitochondrial rRNA methyltransferase 1	–0.46	3.75 × 10^–3^
PVALB	Parvalbumin	–0.46	3.69 × 10^–3^
FBXO40	F-box protein 40	–0.46	3.47 × 10^–3^
SMIM5	Small integral membrane protein 5	–0.46	3.44 × 10^–3^
GFI1B	Growth-factor-independent 1B transcriptional repressor	–0.47	3.24 × 10^–3^
GOLGA8G	Golgin A8 family member G	–0.47	3.03 × 10^–3^
CEP19	Centrosomal protein 19	–0.48	2.47 × 10^–3^
SELENBP1	Selenium-binding protein 1	–0.48	2.32 × 10^–3^
ARHGEF40	Rho guanine nucleotide exchange factor 40	–0.48	2.28 × 10^–3^
TMEM200B	Transmembrane protein 200B	–0.48	2.25 × 10^–3^
ATG9A	Autophagy-related 9A	–0.48	2.23 × 10^–3^
CYP11A1	Cytochrome P450 family 11 subfamily A member 1	–0.48	2.11 × 10^–3^
GRAP2	GRB2-related adaptor protein 2	–0.49	1.99 × 10^–3^
MAP6	Microtubule-associated protein 6	–0.49	1.85 × 10^–3^
CENPK	Centromere protein K	–0.49	1.60 × 10^–3^
MOV10	Mov10 RISC complex RNA helicase	–0.50	1.34 × 10^–3^
DFFB	DNA fragmentation factor subunit β	–0.50	1.32 × 10^–3^
TEF	TEF transcription factor, PAR bZIP family member	–0.51	1.13 × 10^–3^
CNN1	Calponin 1	–0.52	7.64 × 10^–4^
C12orf75	Chromosome 12 open reading frame 75	–0.53	6.80 × 10^–4^
PHOSPHO1	Phosphoethanolamine/phosphocholine phosphatase 1	–0.53	6.34 × 10^–4^
LTO1	LTO1 maturation factor of ABCE1	–0.56	2.55 × 10^–4^
RECQL5	RecQ like helicase 5	–0.56	2.43 × 10^–4^
CD19	CD19 molecule	–0.58	1.34 × 10^–4^

**Table 3 genes-14-02101-t003:** 146 protein-coding genes that showed a correlation with a higher risk of PGD (*p* < 0.005).

Symbol	Description	*p*-Value
SULT2A1	Sulfotransferase family 2A member 1	8.05 × 10^–5^
ZFTA	Zinc finger translocation associated	1.15 × 10^–4^
ITIH3	Inter-α-trypsin inhibitor heavy chain 3	1.94 × 10^–4^
ATAT1	α tubulin acetyltransferase 1	2.82 × 10^–4^
LOC124900286	Putative uncharacterized protein DIP2C-AS1	4.01 × 10^–4^
SLC38A3	Solute carrier family 38 member 3	4.02 × 10^–4^
AHSG	α 2-HS glycoprotein	4.16 × 10^–4^
HAO1	Hydroxyacid oxidase 1	4.25 × 10^–4^
NPBWR1	Neuropeptides B and W receptor 1	4.37 × 10^–4^
OOEP	Oocyte-expressed protein	5.00 × 10^–4^
LOC102724250	Neuroblastoma breakpoint family member 1-like	5.06 × 10^–4^
APOH	Apolipoprotein H	5.07 × 10^–4^
KRT17	Keratin 17	5.68 × 10^–4^
CYP4A11	Cytochrome P450 family 4 subfamily A member 11	6.55 × 10^–4^
MT1H	Metallothionein 1H	6.59 × 10^–4^
FGFR1	Fibroblast growth factor receptor 1	6.66 × 10^–4^
TPSD1	Tryptase delta 1	6.96 × 10^–4^
SLC39A14	Solute carrier family 39 member 14	7.12 × 10^–4^
TAP2	Transporter 2, ATP-binding cassette subfamily B member	7.60 × 10^–4^
CEP295NL	CEP295 N-terminal like	7.60 × 10^–4^
CPB2	Carboxypeptidase B2	7.79 × 10^–4^
CYP3A7	Cytochrome P450 family 3 subfamily A member 7	7.92 × 10^–4^
PRR15	Proline-rich 15	8.70 × 10^–4^
UCP1	Uncoupling protein 1	9.73 × 10^–4^
HOXD3	Homeobox D3	9.81 × 10^–4^
ITIH1	Inter-α-trypsin inhibitor heavy chain 1	9.86 × 10^–4^
RIPOR2	RHO family-interacting cell polarization regulator 2	9.86 × 10^–4^
NBR1	NBR1 autophagy cargo receptor	9.86 × 10^–4^
PTPRU	Protein tyrosine phosphatase receptor type U	9.93 × 10^–4^
PAH	Phenylalanine hydroxylase	9.93 × 10^–4^
SERPIND1	Serpin family D member 1	1.03 × 10^–3^
IGFL4	IGF-like family member 4	1.09 × 10^–3^
TMEM150B	Transmembrane protein 150B	1.10 × 10^–3^
TRIM69	Tripartite motif containing 69	1.12 × 10^–3^
CSK	C-terminal Src kinase	1.12 × 10^–3^
YJU2	YJU2 splicing factor homolog	1.12 × 10^–3^
LRCOL1	Leucine-rich colipase like 1	1.15 × 10^–3^
PRG3	Proteoglycan 3, pro eosinophil major basic protein 2	1.23 × 10^–3^
C2	Complement C2	1.27 × 10^–3^
SEPTIN1	Septin 1	1.27 × 10^–3^
PRSS37	Serine protease 37	1.33 × 10^–3^
UGT2A3	UDP glucuronosyltransferase family 2 member A3	1.33 × 10^–3^
BTG2	BTG anti-proliferation factor 2	1.44 × 10^–3^
HAMP	Hepcidin antimicrobial peptide	1.51 × 10^–3^
CYP2A6	Cytochrome P450 family 2 subfamily A member 6	1.52 × 10^–3^
GPR135	G-protein-coupled receptor 135	1.53 × 10^–3^
FGF23	Fibroblast growth factor 23	1.61 × 10^–3^
PRSS22	Serine protease 22	1.62 × 10^–3^
TXNIP	Thioredoxin-interacting protein	1.62 × 10^–3^
TNK2	Tyrosine kinase non receptor 2	1.62 × 10^–3^
ZNG1C	Zn-regulated GTPase metalloprotein activator 1C	1.62 × 10^–3^
DDX43	DEAD-box helicase 43	1.70 × 10^–3^
LGI2	Leucine-rich repeat LGI family member 2	1.80 × 10^–3^
TNFRSF21	TNF receptor superfamily member 21	1.83 × 10^–3^
ADAR	Adenosine deaminase RNA specific	1.83 × 10^–3^
MATN1	Matrilin 1	1.83 × 10^–3^
IKZF3	IKAROS family zinc finger 3	1.83 × 10^–3^
CSF2RB	Colony-stimulating factor 2 receptor subunit β	1.83 × 10^–3^
UGT1A7	UDP glucuronosyltransferase family 1 member A7	1.84 × 10^–3^
STRC	Stereocilin	1.88 × 10^–3^
MT1G	Metallothionein 1G	1.88 × 10^–3^
CPA5	Carboxypeptidase A5	1.97 × 10^–3^
DCSTAMP	Dendrocyte-expressed seven transmembrane protein	2.03 × 10^–3^
HJV	Hemojuvelin BMP co-receptor	2.05 × 10^–3^
PLCG2	Phospholipase C γ 2	2.06 × 10^–3^
GUCY2D	Guanylate cyclase 2D, retinal	2.06 × 10^–3^
DACT1	Dishevelled-binding antagonist of β catenin 1	2.09 × 10^–3^
SLC10A1	Solute carrier family 10 member 1	2.16 × 10^–3^
FGF17	Fibroblast growth factor 17	2.20 × 10^–3^
SAA4	Serum amyloid A4, constitutive	2.22 × 10^–3^
SLC23A1	Solute carrier family 23 member 1	2.27 × 10^–3^
HERC4	HECT and RLD domain containing E3 ubiquitin protein ligase 4	2.32 × 10^–3^
OR8K1	Olfactory receptor family 8 subfamily K member 1	2.39 × 10^–3^
CHRM4	Cholinergic receptor muscarinic 4	2.47 × 10^–3^
RAB7B	RAB7B, member RAS oncogene family	2.54 × 10^–3^
KCNS2	Potassium voltage-gated channel modifier subfamily S member 2	2.54 × 10^–3^
CYP3A4	Cytochrome P450 family 3 subfamily A member 4	2.55 × 10^–3^
BANK1	B-cell scaffold protein with ankyrin repeats 1	2.56 × 10^–3^
CEL	Carboxyl ester lipase	2.56 × 10^–3^
CRYBG2	Crystallin β-γ domain containing 2	2.61 × 10^–3^
IRF1	Interferon regulatory factor 1	2.61 × 10^–3^
HP	Haptoglobin	2.61 × 10^–3^
TMEM221	Transmembrane protein 221	2.66 × 10^–3^
EPPIN-WFDC6	EPPIN-WFDC6 readthrough	2.67 × 10^–3^
BAAT	Bile acid-CoA:amino acid N-acyltransferase	2.76 × 10^–3^
HS3ST3A1	Heparan sulfate-glucosamine 3-sulfotransferase 3A1	2.79 × 10^–3^
IDUA	α-L-iduronidase	2.81 × 10^–3^
APOA1	Apolipoprotein A1	2.82 × 10^–3^
KIAA1522	KIAA1522	2.92 × 10^–3^
DTX3L	Deltex E3 ubiquitin ligase 3L	2.92 × 10^–3^
ORC3	Origin recognition complex subunit 3	2.92 × 10^–3^
STX17	Syntaxin 17	2.92 × 10^–3^
ANKRD13A	Ankyrin repeat domain 13A	2.92 × 10^–3^
ZSWIM4	Zinc finger SWIM-type containing 4	2.92 × 10^–3^
ITPKA	Inositol-trisphosphate 3-kinase A	2.97 × 10^–3^
AKR7A3	Aldo-keto reductase family 7 member A3	2.98 × 10^–3^
SALL1	Spalt-like transcription factor 1	3.09 × 10^–3^
PCDHAC1	Protocadherin α subfamily C, 1	3.16 × 10^–3^
C9	Complement C9	3.16 × 10^–3^
RGS18	Regulator of G protein signaling 18	3.27 × 10^–3^
DIAPH1	Diaphanous-related formin 1	3.27 × 10^–3^
IGBP1	Immunoglobulin-binding protein 1	3.27 × 10^–3^
TRIM22	Tripartite motif containing 22	3.27 × 10^–3^
L3MBTL1	L3MBTL histone methyl-lysine binding protein 1	3.27 × 10^–3^
ALX4	ALX homeobox 4	3.37 × 10^–3^
CXCL14	C-X-C motif chemokine ligand 14	3.40 × 10^–3^
IFNL3	Interferon lambda 3	3.42 × 10^–3^
SH2D1B	SH2 domain containing 1B	3.46 × 10^–3^
LRRC56	Leucine-rich repeat containing 56	3.46 × 10^–3^
EHD1	EH domain containing 1	3.65 × 10^–3^
SMARCC2	SWI/SNF-related, matrix-associated, actin-dependent regulator of chromatin subfamily c member 2	3.65 × 10^–3^
RDH8	Retinol dehydrogenase 8	3.72 × 10^–3^
ABCA8	ATP-binding cassette subfamily A member 8	3.79 × 10^–3^
UNC93A	unc-93 homolog A	3.82 × 10^–3^
USP21	Ubiquitin-specific peptidase 21	3.82 × 10^–3^
PCDH11Y	Protocadherin 11 Y-linked	3.99 × 10^–3^
HOXB9	Homeobox B9	4.07 × 10^–3^
ZMYM6	Zinc finger MYM-type containing 6	4.08 × 10^–3^
POLR1A	RNA polymerase I subunit A	4.08 × 10^–3^
RETREG1	Reticulophagy regulator 1	4.08 × 10^–3^
AKAP17A	A-kinase-anchoring protein 17A	4.08 × 10^–3^
PDCD4	Programmed cell death 4	4.08 × 10^–3^
DRC3	Dynein-regulatory complex subunit 3	4.08 × 10^–3^
SPOP	Speckle-type BTB/POZ protein	4.08 × 10^–3^
PACSIN2	Protein kinase C and casein kinase substrate in neurons 2	4.08 × 10^–3^
KCNIP4	Potassium voltage-gated channel interacting protein 4	4.14 × 10^–3^
TMEM132C	Transmembrane protein 132C	4.16 × 10^–3^
C15orf48	Chromosome 15 open reading frame 48	4.17 × 10^–3^
BTBD19	BTB domain containing 19	4.20 × 10^–3^
ADAM20	ADAM metallopeptidase domain 20	4.20 × 10^–3^
C8G	Complement C8 γ chain	4.25 × 10^–3^
KRTAP10-8	Keratin-associated protein 10-8	4.27 × 10^–3^
ADAMTS15	ADAM metallopeptidase with thrombospondin type 1 motif 15	4.35 × 10^–3^
KRT71	Keratin 71	4.38 × 10^–3^
TTR	Transthyretin	4.40 × 10^–3^
DDX60L	DExD/H-box 60 like	4.54 × 10^–3^
CD19	CD19 molecule	4.54 × 10^–3^
DPEP2	Dipeptidase 2	4.54 × 10^–3^
SOCS7	Suppressor of cytokine signaling 7	4.54 × 10^–3^
KRT83	Keratin 83	4.59 × 10^–3^
TGM3	Transglutaminase 3	4.62 × 10^–3^
ENTPD8	Ectonucleoside triphosphate diphosphohydrolase 8	4.73 × 10^–3^
TNP2	Transition protein 2	4.84 × 10^–3^
DDIT4L	DNA damage inducible transcript 4 like	4.91 × 10^–3^
F11	Coagulation factor XI	4.98 × 10^–3^
LMX1B	LIM homeobox transcription factor 1 β	4.98 × 10^–3^

**Table 4 genes-14-02101-t004:** Four significant protein-coding genes associated with both TnI and PGD after HTx.

Symbol	Biological Relevance	TnI R	TnI *p*-Value	PGD *p*-Value
ITPKA	Calmodulin-binding, calcium signaling	0.46	3.65 × 10^–3^	2.97 × 10^–3^
	Inositol phosphate metabolism			
	Phosphatidylinositol signaling system			
DDIT4L	Negative regulation of signal transduction	0.54	4.99 × 10^–4^	4.91 × 10^–3^
	Inhibits cell growth via the TOR signaling pathway and downstream of AKT1			
CD19	Normal B-cell differentiation	–0.58	1.34 × 10^–4^	4.54 × 10^–3^
	Activation of PI3K and the mobilization of intracellular Ca^2+^ stores			
	Positive regulation of calcium ion transmembrane transport via BCR signaling pathway			
CYP4A11	Metabolism of drugs, fatty acids, arachidonic acids	0.47	2.94 × 10^–3^	6.55 × 10^–4^
	Biosynthesis of cholesterol, steroids, and other lipids PPAR signaling pathway, atherosclerosis			

## Data Availability

The data of this study are not publicly available due to privacy and ethical restrictions. However, the RNA-seq raw counts and normalized data are provided in the Appendix A.

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
