# Peer review of "Transcriptomic Landscape of Circulating Extracellular Vesicles in Heart Transplant Ischemia–Reperfusion"

_genes, 2023, doi:10.3390/genes14112101_

Round 1
Reviewer 1 Report
Comments and Suggestions for Authors
I considered the manuscript entitled “Transcriptomic Landscape of Circulating Extracellular Vesicles in Heart Transplant Ischemia-Reperfusion” by SeoJeong Joo, et al, that is intended to be published in Genes.
I liked the study because its rigor. It shows that thousands of genes are related in IRI after cardiac transplantation. There is arduous laboratory work with an exhaustive search in the most advanced databases. It gives interesting information of the phenomenon.
Some concerns:
Authors determine four genes which showed a moderate correlation with both cardiomyocyte injury and PGD. They speculate about the potential relationship between gene function and PGD. It may be ok for ITPKA and DDIT4L in the IRI situation. However, it is hard to correlate for the other two genes. CD19 by one hand governs normal B cell differentiation, and CYP4A11 is a classical pathway of metabolism of drugs, fatty acids, arachidonic acids. Although the authors look for collateral functions, the actual association seems weak. What is your true opinion? It is a strongly unknown function of these genes, or it is simply the result of chance.
What’s the meaning of edema in this sentence? “Collectively, our results may help to understand how the cellular structure disruption and inflammation after IRI contribute to the development of edema and PGD after HTx”.
To me, the number of patients is not a limitation of the study thanks to the high complexity of the analysis. But I agree with the other limitation announced: “samples collected after the ischemic period, which might have affected the circulating EV
transcriptome with confounding factors at an earlier timepoint during the operation”. By one hand, two control groups should be introduced. By one hand, patients with cardiac surgery without cardiac IRI as, for instance, aortocoronary bypass with hearts beating. And in the other hand aortocoronary bypass with extracorporeal circulation and cardiac arrest. These two groups will allow to discern whether the findings relate to IRI or alloreactivity.
Finally, results in peripheral blood are far to come only from cardiac sources. In this hemodynamic storm, all the organs suffer and release thousands of mediators Should it be better to collect the blood from coronarious sinus or from right atrium through venous central catether?
Reviewer 2 Report
Comments and Suggestions for Authors
The manuscript by Seo Jeong Joo et al, addresses association of differential expression genes (DEGs) from extracellular vesicles (EVs), as study design to Ischemia-reperfusion injury (IRI). Increasing evidence suggests that after reperfusion (recipient = 12h) the transcript profile from EVs differ of before reperfusion (recipient = 0h), as also between primary graft dysfunctions (PGD) and No-PGD; by using EV isolation technique, RNA sequencing and in silico analysis by bioinformatic tools; highlighting that 4 plasma EV genes (ITPKA, DDIT4L, CD19, and CYP4A11) are correlated with cardiomyocyte injury and PGD. Here, the authors investigated the whole-transcriptome EVs in plasma from 38 heart transplant recipients immediately before and at 12h hours after transplant reperfusion.
Major concerns
The authors study the association of DEGs from EV samples in Ischemia-reperfusion injury (IRI). However, there are some concerns that should be explained by the authors.
1.- According to results obtained by RNA sequencing, the authors identified 1317 DEGs and analyzed the features of functions based on in silico analysis. However, all these genes have differential power of discrimination (12h vs 0h) . The authors should analysis the genes or gene-signatures with high power discrimination for 12h (e.g minimum of sensitivity: 80% and specificity 80%). This analysis could be performed by using CombiROC software (http://combiroc.eu/) or R-package.
2.- In section 3.5, the authors stated a selection of top-50 (up- and down-regulated) based on log fold-change. These genes may not be within of the best gene ontology (GO) term (Figure S1). Why did the authors not consider the genes within the best GO term?
3.- This study shows a discovery analysis based on high throughput technique (RNA-sequencing), the authors should validate their findings (ITPKA, DDIT4L, CD19, and CYP4A11) by real-time PCR in the same samples.
Minor concerns
Material and Methods
In section 2.4.- The authors should detail the RNA-sequencing data. RPKM, FPKM or TPM?
Functional annotations by bioinformatics tools should be in an independent section.
Results
In Figure 2.- the authors should show the EV concentrations for each sample (0h: n=5 and 12h: n=8).
Comments on the Quality of English Language
Minor editing of English language required
Reviewer 3 Report
Comments and Suggestions for Authors
The study by Joo et al. investigates ischemia-reperfusion injury (IRI) in heart transplantation, a significant concern for allograft damage. The research involves RNA sequencing of plasma extracellular vesicles (EV) from 38 heart transplant recipients before and 12 hours after reperfusion. The findings reveal changes in the expression of 1317 protein-coding genes in plasma EV after reperfusion, with upregulated genes linked to metabolism and immune activation and downregulated genes related to cell survival and extracellular matrix organization. The study also establishes correlations between EV gene expression and graft IRI and primary graft dysfunction. Four plasma EV genes are identified as sensitive indicators of reperfusion injury. This study sheds light on the unappreciated role of EVs in heart ischemia reperfusion injury and has the potential to find out useful markers for post-heart transplant graft dysfunction. However, several revisions are suggested to enhance to quality of manuscript.
1. While the study presents encouraging data, a significant limitation lies in the relatively short follow-up duration. It's reasonable that the authors focused on early EV changes and their correlation with primary graft dysfunction (PGD) within the first 24 hours. However, it would be highly beneficial to extend the analysis to examine how these early changes might influence long-term outcomes. This is important because the outcome is also influenced by the severity of ischemia-reperfusion (IR) injury, and understanding the implications of early changes for long-term results could provide more comprehensive insights into the clinical impact of the study's findings.
2. Thirteen out of thirty-eight recipients, constituting one-third of the sample, developed primary graft dysfunction (PGD), which is a relatively high ratio. It would be valuable for the study to provide a more in-depth discussion regarding potential explanations for this higher incidence of PGD.
3. Including information about donor characteristics would indeed be beneficial. Such data could provide valuable insights into the potential factors contributing to the high incidence of PGD.
4. In Figure 4E, the authors explored various tissue-cell-specific signature databases. It would be valuable to include information about immune cells and whether EVs originate from different types of immune cells.
5. To enhance the study's robustness, it would be advantageous to validate the four protein-coding genes (ITPKA, DDIT4L, CD19, and CYP4A11) at the protein level.
6. Given the connection between gene transcripts in circulating EVs and allograft rejection at 12 hours after reperfusion, it would be helpful to obtain more follow-up data on confirmed rejection cases. This additional information would enhance the study's prognostic value by demonstrating the predictive power of these EV genes in identifying and potentially preventing allograft rejection over an extended period.
7. It is important to carefully review and verify the data presented in the tables. There appears to be a potential error with the recipient age data in Table 1.
Comments on the Quality of English Languageminor editing
Round 2
Reviewer 1 Report
Comments and Suggestions for Authors
none
Reviewer 2 Report
Comments and Suggestions for Authors
In this new version, the authors have addressed the major concerns raised by the reviewer.